# Ultrafast Dynamics of Valley-Polarized Excitons in WSe_2_ Monolayer Studied by Few-Cycle Laser Pulses

**DOI:** 10.3390/nano13071207

**Published:** 2023-03-28

**Authors:** Petr Koutenský, Artur Slobodeniuk, Miroslav Bartoš, František Trojánek, Petr Malý, Martin Kozák

**Affiliations:** 1Department of Chemical Physics and Optics, Faculty of Mathematics and Physics, Charles University, Ke Karlovu 3, 121 16 Prague, Czech Republic; 2Department of Condensed Matter Physics, Faculty of Mathematics and Physics, Charles University, Ke Karlovu 5, 121 16 Prague, Czech Republic; 3Central European Institute of Technology, Brno University of Technology, Purkyňova 656/123, 612 00 Brno, Czech Republic

**Keywords:** transition metal dichalcogenide monolayers, femtosecond transient reflection spectroscopy, intravalley excitons dynamics

## Abstract

We report on the experimental investigation of the ultrafast dynamics of valley-polarized excitons in monolayer WSe${}_{2}$ using transient reflection spectroscopy with few-cycle laser pulses with 7 fs duration. We observe that at room temperature, the anisotropic valley population of excitons decays on two different timescales. The shorter decay time of approximately 120 fs is related to the initial hot exciton relaxation related to the fast direct recombination of excitons from the radiative zone, while the slower picosecond dynamics corresponds to valley depolarization induced by Coloumb exchange-driven transitions of excitons between two inequivalent valleys.

## 1. Introduction

Valleytronics in transition metal dichalcogenide monolayers (2D TMDs) has become a widely studied topic aiming for the development of a new type of electronics, in which the role of electron charge is replaced by a new quantum number associated with the specific band structure extrema occupied by an exciton [1,2,3,4,5,6,7]. Two-dimensional TMDs seem to be ideal for such application, thanks to a large spin-orbit splitting [8,9], leading to inequivalent groups of energy-degenerate extrema (valleys) in their band structure with different pseudospin of the lowest state and thanks to the specific optical selection rules allowing the excitation and read-out of valley-polarized excitons using circularly polarized light [7,8,10,11]. Excitons in these materials have also very strong coupling with photons and high binding energy due to the large effective masses of charge carries [12,13,14]. On the other hand, practical valleytronic applications of 2D TMDs are hindered by a short valley polarization lifetime caused most probably by the coupling of the excitons in different valleys by Coulomb exchange interaction [15,16,17,18,19]. The experimental values of valley lifetime at room temperature are limited to several picoseconds or shorter [20,21,22]. While the picosecond exciton dynamics in these materials has been studied both by pump-probe measurements and time-resolved photoluminescence spectroscopy, the initial stage of the exciton and valley polarization dynamics has not been investigated in detail [23,24,25,26]. After the coherent optical excitation, the excited carrier distribution relaxes via electron–electron and electron–phonon interaction. The typical timescales of the relaxation processes depend on the excess energy of carriers, carrier density, and the strength of the electron–phonon interaction and can be as short as a few tens to hundreds of femtoseconds [10]. Such time resolution is achievable using time-resolved optical spectroscopy with ultrashort laser pulses with a pulse duration of several femtoseconds.

Here, we study the valley polarization of excitons in monolayered WSe${}_{2}$ using transient reflectivity measurements with circularly polarized few-cycle laser pulses. We generate and measure the anisotropy of valley populations by using different combinations of handedness of circularly polarized pump and probe beams with time resolution of 13 fs. We show that the valley dynamics have two components with different origins. The fast component is related to valley depolarization due to carrier relaxation within the first 120 fs after the excitation, while the second picosecond component corresponds to the valley lifetime of excitons at room temperature.

## 2. Materials and Methods

The ultrafast dynamics of valley polarization of excitons in monolayer ${\mathrm{WSe}}_{2}$ is studied using degenerate non-collinear transient reflection spectroscopy (pump–probe technique) with circularly polarized ultrashort pulses. The spectrum of the probe beam reflected from the sample is measured as a function of the time delay between the pump and probe pulses. The layout of the experimental setup is shown in Figure 1. Few-cycle pulses are generated in a Ti:sapphire laser oscillator Rainbow (Femtolasers); their duration is approximately 7 fs (FWHM, measured by spectral phase interferometry for direct electric field reconstruction) with a spectrum spanning the region 650–950 nm (1.3–1.9 eV). The temporal response of the setup is characterized using sum–frequency cross-correlation in a thin BBO crystal. The resulting response function has FWHM of 13 fs, and it is plotted as a blue dotted curve in Figure A2. The repetition rate of the laser is 75 MHz and the maximum pulse energy of the pump pulse is 4 nJ. The group delay dispersion of the optical elements of the setup is compensated using chirped mirrors (DCM7, Venteon) and fused silica wedges to compress the pulses on the sample. The broad spectrum of the pulses covers the photon energy region, which includes also the studied 1sA exciton resonance in ${\mathrm{WSe}}_{2}$ monolayer. A silver parabolic mirror (f = 5 cm) is used to focus the pump and probe laser beams onto the sample. The 1/e${}^{2}$ radii of the pump and probe beams on the sample are 20 μm and 10 μm, respectively. The full spectra of the probe pulse are measured as a function of the time delay between the pump and probe pulses using a grating spectrograph (Shamrock 163) and a cooled CCD camera (Andor iDUS 420). The transient reflectivity spectra are calculated by dividing the spectra of the probe pulse incident on a sample in a certain time delay after the pump pulse (excited sample) by the spectrum of the probe reflected from the sample before the pump pulse (unexcited sample). The signal amplitude of few percent allows to use a slow camera without an optical chopper. The circular polarization is generated using broadband quarter-wave plates. To compensate for the phase shift between the s- and p-polarization components obtained due to the reflections at mirrors placed between the waveplate and the sample, we use a periscope setup (cross in Figure 1) with two additional reflections in the vertical direction. The resulting quality of circular polarization is characterized using a quality factor defined using the intensities of the two counter-rotating field components $Q=\left({\left|E_{\sigma^{+}}\right|}^{2}-{\left|E_{\sigma^{-}}\right|}^{2}\right)/\left({\left|E_{\sigma^{+}}\right|}^{2}+{\left|E_{\sigma^{-}}\right|}^{2}\right)$. The measured values of the quality factor of the pump and probe beams are $Q_{\mathrm{pump}}=96$% and $Q_{\mathrm{probe}}=95$%, respectively.

The WSe${}_{2}$ monolayers are prepared by gel-film-assisted mechanical exfoliation from bulk WSe${}_{2}$ crystal [27,28]. The first step in the sample fabrication is the separation of clear surface crystals from the bulk crystal using standard dicing tape (F05). The material is then moved from the dicing tape to the gel film by pressing and pulling away the tape. In this step, only the top few layers are typically attached to the gel film. Here, we search for the monolayer using an optical microscope and transfer it to the Si/SiO${}_{2}$ substrate. A similar procedure is performed with hBN multilayers, which are deterministically transferred to the same location on the substrate and cover the WSe${}_{2}$ monolayer. The exfoliated WSe${}_{2}$ monolayers are transferred to a Si/SiO${}_{2}$ substrate consisting of bulk crystalline silicon and a 90 nm thick layer of SiO${}_{2}$ at the surface. The monolayers are covered by multilayered hBN to protect them from degradation in the ambient atmosphere. The size of the monolayer flake is 20 μm, which is larger than the size of the probe beam. The basic characterization of the monolayer is performed using differential reflectivity measurement at room temperature, which reveals a characteristic 1sA exciton peak at 1.639 eV (see [29]) and by photoluminescence spectroscopy, which confirms the position of the exciton resonance at the energy corresponding to monolayered WSe${}_{2}$ (the measured photoluminescence spectrum is shown in Figure A1). Note that the encapsulation of the monolayer with hBN from both sides, in general, provides the different shape of the optical response [30].

## 3. Results and Discussion

The measured dynamics of transient reflectivity spectra $\Delta R/R_{0}\left(t,\hbar \omega \right)$ are shown in Figure 2a. $\Delta R/R_{0}\left(t,\hbar \omega \right)$ is defined using the spectrum of the probe reflected from the sample in time delay *t* after the excitation S(t,ℏω) and the reflected spectrum measured with the negative time delay between pump and probe pulses S(−∞,ℏω) as
(1)$$\Delta R/R_{0}\left(t,\hbar \omega \right)\equiv \frac{S(t,\hbar \omega )-S(-\infty ,\hbar \omega )}{S(-\infty ,\hbar \omega )}.$$

The measurements are performed for co-rotating circular polarizations of the pump and probe pulses labeled as $\sigma_{\mathrm{pump}}^{+}/\sigma_{\mathrm{probe}}^{+}$ (upper panel in Figure 2a) and for counter-rotating circular polarizations labeled as $\sigma_{\mathrm{pump}}^{+}/\sigma_{\mathrm{probe}}^{-}$ (lower panel in Figure 2a). In Appendix B, we show that the results are consistent with the measurement with the other two possible combinations of circular polarizations of the pulses $\sigma_{\mathrm{pump}}^{-}/\sigma_{\mathrm{probe}}^{-}$ and $\sigma_{\mathrm{pump}}^{-}/\sigma_{\mathrm{probe}}^{+}$, thus excluding the possible influence of the observed difference between the two signals shown in Figure 2a by the experimental setup (e.g., a possible shift of the beams on the sample due to the rotation of the waveplates when changing the handedness of circular polarization). These features can be understood from the optical properties of the monolayer depicted in Figure A3 in Appendix C.

We observe an increase in reflectivity in the spectral region from 1.61 to 1.67 eV after the arrival of the pump pulse, which is present for both combinations of circular polarizations. This increase in reflectivity corresponds to absorption bleaching of the 1sA exciton resonance with the transition energy of 1.639 eV [31,32] due to the excited population of excitons. The fact that the decrease in absorption in the monolayer leads to an increase in transient reflectivity is caused by the substrate used in our experiments, which is not transparent for light at this photon energy. As shown in [33], when the dielectric function of the substrate has a nonzero complex part, the differential reflectivity may change its sign. This is also confirmed by the differential reflectivity measurements of our samples [29].

The dynamics of the transient reflectivity change integrated over the spectral region of 1.61–1.67 eV for both combinations of circular polarizations is shown in Figure 2b. The dynamics for $\sigma_{\mathrm{pump}}^{+}/\sigma_{\mathrm{probe}}^{+}$ and $\sigma_{\mathrm{pump}}^{+}/\sigma_{\mathrm{probe}}^{-}$ differ mainly in the initial region, but there is a small difference remaining also at the picosecond time scales. For the co-rotating polarizations, we excite and probe the population of the same group of equivalent valleys (e.g., K${}^{+}$ valleys), while for the counter-rotating case, we pump the excitons preferentially in one group of valleys (K${}^{+}$ valleys) and probe the population in the inequivalent group (K${}^{-}$ valleys). The difference between the two signals is thus a sign of transient anisotropy of valley populations in K${}^{+}$ and K${}^{-}$ valleys of the band structure.

In the transient reflectivity data shown in the upper panel of Figure 2a, we can see sharp features, which are present due to the interference between the probe field and a weak scattered field from the pump pulse. The reason why the interference is present only in the $\sigma_{\mathrm{pump}}^{+}/\sigma_{\mathrm{probe}}^{+}$ and not in the $\sigma_{\mathrm{pump}}^{+}/\sigma_{\mathrm{probe}}^{-}$ measurement is the fact that for the opposite handedness of circular polarizations of two waves, the total light intensity can be expressed just as a sum of the two intensities because the interference is suppressed by the opposite phase shifts between the two orthogonal components of the field for right and left-handed polarizations. However, spectral interference does not influence the measured dynamics because it is removed when the spectra are integrated over the region of the exciton resonance.

To study the dynamics of valley polarization of excitons, we subtract the transient reflectivity dynamics of the exciton resonance measured with a counter-rotating combination of circular polarizations from the data with a co-rotating combination. The resulting dynamics of valley polarization are plotted in Figure 3 for five values of the pump fluence on the sample. The dynamics can be successfully modeled using a double-exponential decay function (see the dashed curves fitted to the data in Figure 3) with two time constants $\tau_{1}$ corresponding to the fast femtosecond dynamics and $\tau_{2}$ related to slower picosecond dynamics. We observe that the shorter decay time of $\tau_{1}=\left(116\pm 7\right)$ fs does not depend on the excitation fluence. In contrast, the longer decay time $\tau_{2}$ decreases with increasing pump fluence starting from $\tau_{2}=\left(6\pm 1\right)$ ps at 18 μJ/cm${}^{2}$ to $\tau_{2}=\left(1.1\pm 0.4\right)$ ps at pump fluence of 65 μJ/cm${}^{2}$. The values of the two time constants $\tau_{1}$ and $\tau_{2}$ are plotted in Figure 4a as a function of the pump fluence.

The different dependence of the two dynamics on the fluence of the pump beam points out their different physical origins. The initial fast decay dynamics, which does not depend on pump fluence, is probably related to the time in which the excitonic system relaxes and loses the excess energy, which is a consequence of the broad spectrum of the excitation (up to 1.9 eV). However, the spectrum does not reach the band gap of the material of 2.05 eV and we thus do not expect excitation of free electrons and holes but rather the direct excitation of the 1sA exciton states. Even with above band gap excitation, the excitons are formed within the first few tens of femtoseconds after the excitation [34] due to their high binding energy of about 300 meV. The time scale is significantly longer only for highly nonresonant excitation and high carrier densities [35]. After the excitation, the exciton distribution first reaches the thermal equilibrium due to the exciton–exciton interactions leading to an effective temperature well above the lattice temperature. At the same time, the initial excitonic distribution in the K${}^{+}$ valley is modified via electron–phonon [36] and exciton–exciton [37,38] interactions, and it is also reduced due to the recombination of excitons through the radiative zone. Additionally, the excitons from the K${}^{+}$ valley transfer a part of their population to the K${}^{-}$ valley due to the long-range exchange interaction [18,39]. We assume that the fast signal with a decay time of about 120 fs corresponds to this initial relaxation dynamics. We note that a similar time constant was observed in the measurements of exciton valley coherence time in WSe${}_{2}$ [19].

The slower component of the valley depolarization dynamics corresponds to the recombination of excitons with large momenta, i.e., outside the radiative zone, after their relaxation from the non-radiative to the radiative zone. The dependence on the excitation fluence is caused by the multiparticle processes, such as Auger recombination, which accelerates the recombination dynamics at carrier densities above 4 × 10${}^{9}$ cm${}^{-2}$ [40]. From the known fluence of the pump pulse, we can estimate the exciton density corresponding to the highest fluence to be approximately $5\;\times \;10^{12}$ cm${}^{-2}$. Auger recombination was previously observed in this region of carrier densities [41]. The high-density excitation regime is further supported by the observed sub-linear dependence of the peak of the transient reflectivity signal on the pump fluence *F*, which is caused by state filling. The measured maximum transient reflectivity change as a function of the pump fluence is shown in Figure 4b along with the fits using a saturation function $\Delta R/R_{0}=\left(AF\right)/\left(1+F/F_{\mathrm{sat}}\right)$, where *A* is a constant describing the dependence of the reflectivity change with the fluence in the linear regime and $F_{\mathrm{sat}}$ is the saturation fluence. This simple model can be applied in the monolayer because of the small change of light intensity when propagating through the monolayer. The values of the parameters from the two fits are $A=1.77\times 10^{-5}\;{\mathrm{cm}}^{2}/\mu J$ and $F_{\mathrm{sat}}=84\;\mu J/{\mathrm{cm}}^{2}$ for co-polarized pulses and $A=9.01\times 10^{-6}\;{\mathrm{cm}}^{2}/\mu J$ and $F_{\mathrm{sat}}=101\;\mu J/{\mathrm{cm}}^{2}$ for counter-rotating polarizations.

The initial ultrafast exciton dynamics is characterized more in detail by measuring the transient reflectivity of a linearly polarized probe after the excitation by cross-polarized pump pulse (linear polarization perpendicular to that of the probe). We chose this combination of linear polarizations to excite and probe both inequivalent groups of valleys and at the same time to suppress the interference in the transient reflectivity signal caused by scattering of the pump in the case of parallel polarizations of both beams. The measured data are shown in Figure 5a. We observed that the initial exciton formation is very fast with the growth time of the peak corresponding to the 1sA exciton resonance of 110 fs (time between zero and maximum transient reflectivity change). We also observed a shift of the spectra and a decrease of transient reflectivity at photon energies below and above the resonance. The changes of the reflectivity spectrum as a function of the time delay are shown in Figure 5b. Here, we see that the growth of the exciton peak is accompanied with a slight spectral shift and that the negative transient reflectivity signal is present only during the first 200 fs.

Besides the initial exciton relaxation, the negative signal and ultrafast peak shift is influenced by the presence of coherent optical phenomena. Specifically, the optical Stark shift has been widely studied in the 2D materials [42,43,44] and has been shown to shift the exciton resonance during the time in which the sample is illuminated by the pump light. The optical Stark effect can be theoretically explained semiclassically with two-level approximation [45] or using more advanced theory utilizing semiconductor Bloch equations [29]. In general, the energy shift of the exciton level is approximately indirectly proportional to the detuning $\Delta =E_{0}-\hbar \omega_{p}$ between the transition energy $E_{0}$ and the pump photon energy $\hbar \omega_{p}$. In the previous measurements, the spectral width of the pump pulses was typically smaller or comparable to the exciton linewidth, allowing to specify the detuning energy. In our case of few-cycle pulses, the pump spectrum is much broader than the exciton line. Thus, the classical description of the coherent optical phenomena with the constant envelope of the oscillating field is not valid anymore. The oscillatory behavior of the transient absorption of exciton resonance induced by a resonant and off-resonant ultrashort pump pulse has been theoretically predicted in quantum wells [46].

In order to provide a more detailed picture of the decay mechanisms of the transient reflectivity of the TMD monolayer under a few-femtosecond pump pulse, we consider the experimentally measured value $\Delta R/R_{0}$, integrated over the energy domain in between $E_{1}=1.61$ eV and $E_{2}=1.67$ eV. Namely, we introduce the following objects: ${\left[\Delta R/R_{0}\right]}^{++}\equiv \int_{E_{1}}^{E_{2}}dE\left[\Delta R/R_{0}\right]$ for the $\sigma_{\mathrm{pump}}^{+}/\sigma_{\mathrm{probe}}^{+}$ case, and ${\left[\Delta R/R_{0}\right]}^{+-}\equiv \int_{E_{1}}^{E_{2}}dE\left[\Delta R/R_{0}\right]$ for the $\sigma_{\mathrm{pump}}^{+}/\sigma_{\mathrm{probe}}^{-}$ case. Both of them are presented in Figure 2b as a function of time delay. One can observe that ${\left[\Delta R/R_{0}\right]}^{++}$ behaves differently than ${\left[\Delta R/R_{0}\right]}^{+-}$. Namely, the first one decays faster than the second one.

We suppose that the increase in the transient reflectivity of the sample can be explained by the Pauli blocking phenomena—the larger the number of quasiparticles, that form excitons in the TMD monolayer, that occupy the valence and conduction bands, the less space remains for the new optical transition, i.e., for the absorption of incoming light. Note that the pump pulse in a frequency domain covers the energy region from 1.35 eV up to 1.9 eV. The upper photon energy of the pump pulse is lower than the bandgap of the monolayer $E_{\mathrm{bg}}\approx 2.02$ eV [29], and hence we conclude that the pump pulse generates mainly the excitonic states in the monolayer. Therefore, the value of the reflectivity in K${}^{+}$ valley at some energy *E* can be associated with the occupation number $n_{+}\left(E,t\right)$ of the excitons generated in this valley by $\sigma^{+}$ polarized pump pulse. This statement is in agreement with the result in Figure 4b, where the transient reflectivity of the sample increases with the increase in the pump fluence since more fluence creates more excitons. Note that the transient reflectivity also increases in the valley, opposite to the one where the optical transitions happen, i.e., in the case $\sigma_{\mathrm{pump}}^{+}/\sigma_{\mathrm{probe}}^{-}$ represented by the curve in Figure 2b). Such an effect indicates the appearance of a non-zero value of the occupation number of the excitons $n_{-}\left(E,t\right)$ in the valley (K${}^{-}$) opposite to the one (K${}^{+}$) where optical transitions happen. In order to describe simultaneously the excitons’ dynamics in both valleys, we use the 2×2 density matrix $\rho \left(q,t\right)=\sum_{\xi ,\xi^{\prime }}\rho_{\xi \xi^{\prime }}\left(t\right)|q,\xi \rangle \langle q,\xi^{\prime }|$ in the exciton valley space represented by valley indices $\xi ,\xi^{\prime }=\pm 1$ at momentum q and time *t*. Here, we introduce the exciton state |q,ξ〉 with momentum q in the valley ξ. The diagonal terms of ρ(q,t) stand for the exciton populations $n_{+}\left(q,t\right)$ and $n_{-}\left(q,t\right)$ in valleys K${}^{+}$ and K${}^{-}$ respectively, while the off-diagonal terms describe the coherence between K${}^{+}$ and K${}^{-}$ exciton states wit momentum q at time *t*. We present the density matrix in the form $\rho \left(q,t\right)=N\left(q,t\right)I/2+\sum_{j=x,y,z}S_{j}\left(q,t\right)\sigma_{j}/2$. Here, I is a unit 2×2 matrix and $\sigma_{j}$ are Pauli matrices. Hence, N(q,t)=Tr[ρ(q,t)] and $S_{j}=\mathrm{Tr}\left[\rho \left(q,t\right)\sigma_{j}\right]$. One can see that $N\left(q,t\right)=n_{+}\left(q,t\right)+n_{-}\left(q,t\right)$ is the total population of the excitons with momentum q at the time *t*, i.e., a sum of the populations $n_{+}\left(q,t\right)$ and $n_{-}\left(q,t\right)$ in K${}^{+}$ and K${}^{-}$ valleys, respectively. The parameters $S_{j}\left(q,t\right)$ form so-called valley pseudo-spin $S=(S_{x},S_{y},S_{z})$. Note that the *z* component of this pseudo-spin defines the difference between exciton populations with momentum *q* and time *t* in K${}^{+}$ and K${}^{-}$ valleys of the monolayer, i.e., $S_{z}\left(q,t\right)=n_{+}\left(q,t\right)-n_{-}\left(q,t\right)$. Therefore, the experimental difference ${\left[\Delta R/R_{0}\right]}^{++}-{\left[\Delta R/R_{0}\right]}^{+-}$ can be expressed as $\int_{E_{1}}^{E_{2}}dE\left[n_{+}\left(q\left(E\right),t\right)-n_{-}\left(q\left(E\right),t\right)\right]=\int_{q(E_{1})}^{q(E_{2})}d^{2}qS_{z}\left(q,t\right)$. In other words, the *z* component of the pseudospin provides qualitatively the time dependence of the measured ${\left[\Delta R/R_{0}\right]}^{++}-{\left[\Delta R/R_{0}\right]}^{+-}$ value. The equation of motion for the pseudo-spin vector S(q,t)≡S(E(q),t) (Maialle–Silva–Sham equation) [18,47] reads
(2)$$\begin{array}{c} \frac{\partial S(q,t)}{\partial t}=\Omega \left(q\right)\times S\left(q,t\right)+\sum_{q^{\prime }}\left[W_{{\mathrm{qq}}^{\prime }}S\left(q^{\prime },t\right)-W_{q^{\prime }q}S\left(q,t\right)\right]-\frac{S(q,t)}{\tau }. \end{array}$$

The first term in the right-hand side represents the admixture of the excitonic states with momentum q between K${}^{+}$ and K${}^{-}$ valleys. This term originates from the long-range exchange interaction in the TMD monolayers. The frequency parameter Ω(q) is an in-plane vector (Ω(q)cos(2θ),Ω(q)sin(2θ),0), where $\Omega \left(q\right)=c\Gamma_{0}q\left(n+1\right)/\omega_{0}\left(n^{2}+1\right)$ and θ is an angle between the *x* axis and exciton’s momentum q [39]. Here, $\Gamma_{0}$ is the radiative decay rate of an exciton in the monolayer, *c* is the speed of light, $\omega_{0}$ is the exciton energy in frequency units, q=|q|, and *n* is the refractive index of the substrate. The second term is the collision integral describing the pseudo-spin distribution function in the process of scattering. The parameter $W_{{\mathrm{qq}}^{\prime }}$ describes the rate of scattering of the exciton from the state with the momentum $q^{\prime }$ to the state with the momentum q. Its form is defined by the properties of the particular interaction, e.g., exciton–phonon and/or exciton–exciton interaction, and the scattering by impurities (which induce various effects, e.g., photogate effect [48]). The last term of the equation describes the processes of the destruction of the exciton states with momentum q, e.g., radiative exciton recombination. The corresponding decay time τ is different for each component $S_{j}\left(q,t\right)$.

The dynamics of the pseudospin S is defined by the characteristic times of acting of each term. The characteristic time of the first term can be evaluated as $\tau_{\Omega }=2\pi /\Omega \left(q_{T}\right)$. Here, we choose the exciton’s momentum value $q_{T}$ for the case of the thermalized excitons $\hbar^{2}q_{T}^{2}/2M=k_{B}T$. *M* is an exciton mass, $k_{B}$ is the Boltzmann constant, *T* is a temperature, and *ℏ* is the Plank constant. Considering the room temperature T≈300 K case, the exciton’s mass *M* equal to an electron mass [49], n≈1.45, $\hbar \omega_{0}\approx 1.63$ eV, theoretically estimated $\Gamma_{0}\approx {\left(200\;\mathrm{fs}\right)}^{-1}=7$ meV [50], we obtain $\tau_{\Omega }\approx 7.5$ fs. For the case of the experimentally reported $\Gamma_{0}∼{\left(1\;\mathrm{ps}\right)}^{-1}$ [20,21,22,23,24], we obtain $\tau_{\Omega }\approx 40$ fs. It gives us the characteristic time of the admixture of the excitonic states in between the two valleys. The effectiveness of this mechanism can be checked by considering the equation of motion without scattering and radiative terms. Then, the equation of motion can be solved for each particular momentum q. Let us consider the initial occupation numbers of excitons $n_{+}\left(q,0\right)=f\left(q\right)$ and $n_{-}\left(q,0\right)=0$ generated in K${}^{+}$ and K${}^{-}$ valleys of the monolayer after the pump pulse application in our experiment. The f(q) describes the amount of excitons with the absolute value of momentum *q* in the K${}^{+}$ valley of the monolayer. The shape of this function depends on the properties of the studied material and the geometry of the experiment, e.g., on the profile of the pump pulse in the frequency domain, due to the long-range Coulomb-induced exchange interaction excitonic states with momentum q admix between K${}^{+}$ and K${}^{-}$ valleys with momentum-dependent frequency Ω(q), which leads to the depletion of the K${}^{+}$ valley, and population of the K${}^{-}$ valley, respectively. The difference between total populations in these valleys can be obtained by averaging by the energy (or equivalently by the momentum of excitons) of the *z*-component of the pseudospin $S_{z}\left(q,t\right)$: $\int d^{2}qS_{z}\left(q,t\right)=\int dqqf\left(q\right)\cos\left[\Omega \left(q\right)t\right]$ [39]. For the case of large time *t*, the number of the excitons in the K${}^{+}$ and K${}^{-}$ valleys equilibrates. Note that the admixture of the states becomes more effective for the large momentum *q* of excitons.

The characteristic time of the scattering processes strongly depends on the type of scattering. Using the expression for the second term in the equation of motion, this time can be estimated as a full escaping time of the exciton from the state q to all other states $\tau_{W}^{-1}\left(q\right)=\sum_{q^{\prime }}W_{{\mathrm{qq}}^{\prime }}$. For example, for the scattering of excitons on acoustic phonons, this time can be presented as $\tau_{W}^{-1}\approx A\left(k_{B}T/\hbar \right)$ (see [51] for details). The theoretical estimation of this time for WSe${}_{2}$ monolayer gives A≈0.08 and $\tau_{W}\approx 300$ fs. The experimental values of the parameter A∼1 [21,24,52,53,54] give $\tau_{W}\approx 25$ fs. Finally, the last time parameter of the Maialle–Silva–Sham equation τ≈190 fs was reported in [19]. As one can see, all the characteristic times lie in the femtosecond time domain, which makes the quantitative analysis of the equation of motion hard. The full quantitative analysis can be performed numerically in the frameworks of semiconductor Bloch equations [37,38,55]. It requires knowledge of the many details of the exciton–exciton and exciton–phonon interactions as well as the interaction of the excitons with impurities in the TMD monolayer; that is, however, beyond the current study.

Therefore, we restrict ourselves to the qualitative analysis of the processes presented in the Maialle–Silva–Sham equation. The application of the strong pump pulse generates excitons in K${}^{+}$ valley with their momentum q localized in the radiative zone $q<q_{rad}=\omega_{0}\left(n+1\right)/2c$. The part of these excitons is moved out from the radiative zone by scattering terms $W_{{\mathrm{qq}}^{\prime }}$. At the same time, the excitonic states move from the $K^{+}$ to the K${}^{-}$ valley due to Coulomb exchange terms. The moved excitons give rise to the population $n_{-}\left(q,t\right)$, which is responsible for the increase in the reflectance in the K${}^{-}$ valley (see Figure 2). Finally, the remaining excitons recombine in the radiative zone. The excitons in this zone have a small momentum, and therefore, they expect a small admixture between the valleys. It allows us to neglect this term in the equation of motion. Supposing that the occupation of the non-radiative states is smaller than in a radiative zone, assume that the number of excitonic states coming back to the radiative zone from the non-radiative is small and neglects the corresponding term in the equation of motion. Therefore, for the states in the radiative zone, we can write the simplified equation $\partial S\left(q,t\right)/\partial t\approx -\left[\tau^{-1}+\tau_{W}^{-1}\left(q\right)\right]S\left(q,t\right)$. It describes the fast exponential decay of the excitons from the radiative zone leading to its depletion. We suppose that this process corresponds to the fast decay of the transient reflectivity ${\left[\Delta R/R_{0}\right]}^{++}$, see black curve in the Figure 2b. One can observe that it does not depend on pump fluence, i.e., concentration of the excitons in the monolayer. Such a feature is observed in the experiment. The excitons outside the radiative zone cannot decay radiatively. The only way for their decay is to be scattered back to the radiative zone. We suppose that this process leads to the long tail of the $S_{z}\left(t\right)$ function, which can be approximated by an exponent with a larger decay time. Note that the back-scattering dynamics should depend on the number of excitons outside the radiative zone, manifesting into fluence dependence of the decay time. Such a dependence is also observed in the experiment.

## 4. Conclusions

In summary, we show that the excitonic valley polarization dynamics in monolayer WSe${}_{2}$ visualized using transient reflectivity measurements with few-cycle laser pulses has two components. The fast component with an exponential decay time of 120 fs is related to the initial exciton relaxation dynamics, while the second picosecond component corresponds to the decay of the exciton population. We also show the experimental data of the initial dynamics of transient reflectivity spectra demonstrating the ultrafast evolution on the time scales of a few tens of femtoseconds. Here, we observe the interplay between coherent optical phenomena induced by the pump pulse and the resonant response of the exciton system to the ultrafast perturbation, which is much shorter than the exciton coherence time.

## Figures and Tables

**Figure 1 nanomaterials-13-01207-f001:**
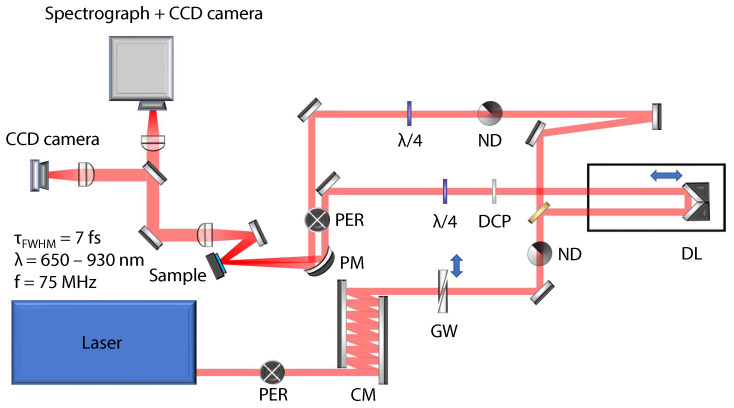
Detailed layout of the experimental setup. Laser—Ti:sapphire laser oscillator Rainbow (Femtolasers/Spectra Physics), λ/4—achromatic quarter-wave plate, PER—periscope, CM—chirped mirrors, GW—glass wedges, ND—neutral densit filter, DCP—dispersion compensating plate, DL—delay line, PM—parabolic mirror.

**Figure 2 nanomaterials-13-01207-f002:**
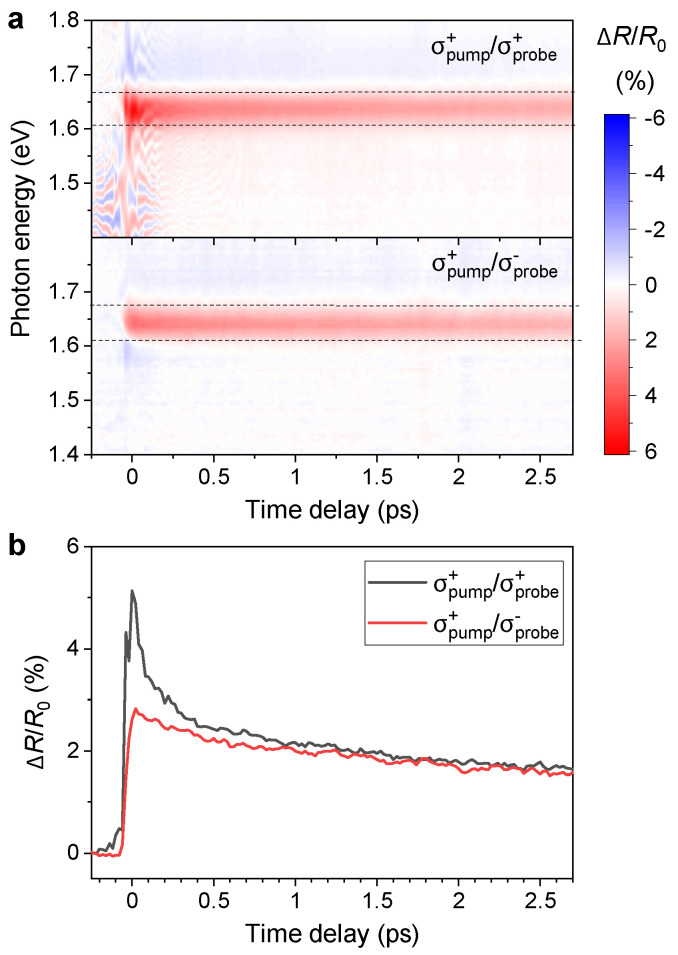
(**a**) Evolution of transient reflectivity for $\sigma_{\mathrm{pump}}^{+}/\sigma_{\mathrm{probe}}^{+}$ combination of circular polarizations of pulses (upper panel) compared to an opposite combination of polarizations $\sigma_{\mathrm{pump}}^{+}/\sigma_{\mathrm{probe}}^{-}$ (lower panel). Both measurements are performed with the pump fluence of 65 μJ/cm${}^{2}$. (**b**) Time evolution of the transient reflectivity spectrally integrated over the exciton resonance (region marked by dashed lines in (**a**)).

**Figure 3 nanomaterials-13-01207-f003:**
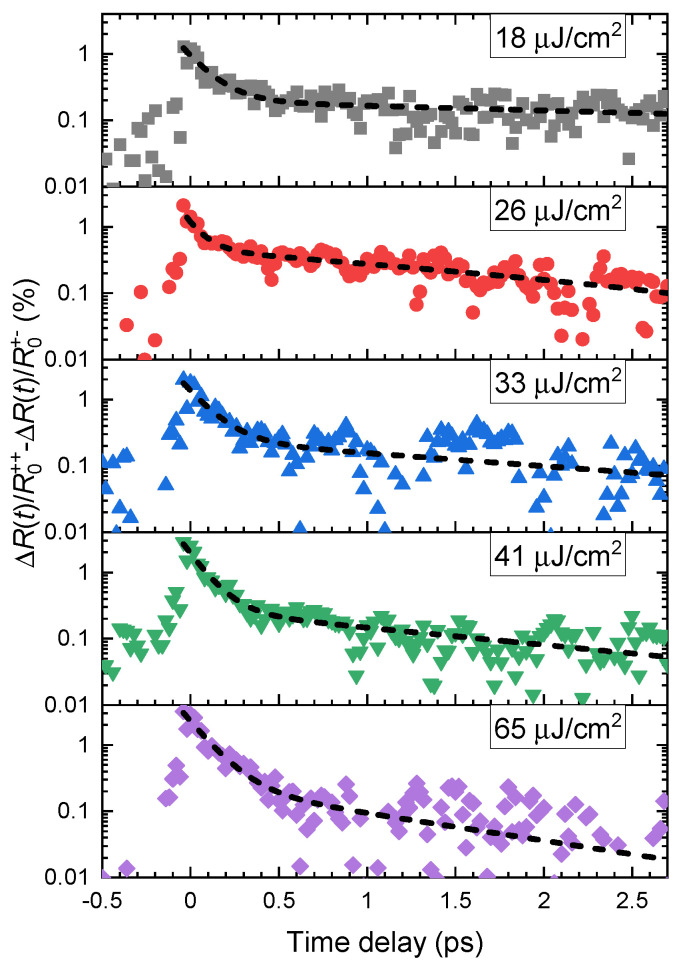
Valley polarization dynamics of 1sA excitons in WSe${}_{2}$ monolayer. The data are obtained by subtracting the transient reflectivity dynamics of the exciton line for counter-rotating polarizations of pump and probe pulses (red curve shown in Figure 2b) from the dynamics measured with co-rotating polarizations (black curve in Figure 2b). The data are spectrally integrated in the photon energy region 1.61–1.67 eV. The dashed curves correspond to double-exponential fitting functions.

**Figure 4 nanomaterials-13-01207-f004:**
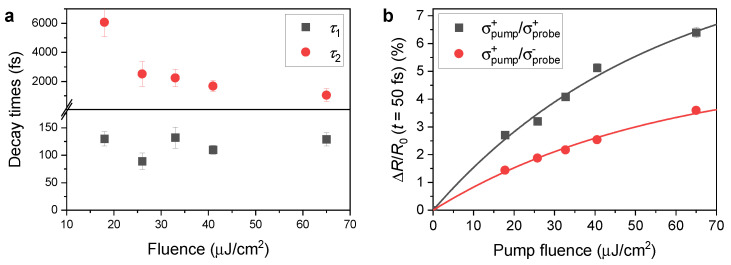
(**a**) Decay times $\tau_{1}$ and $\tau_{2}$ from the double exponential fits of the valley polarization dynamics shown as dashed curves in Figure 3. (**b**) The maxima of the transient reflectivity signal as a function of the pump fluence for both combinations of circular polarizations of pump and probe pulses. Curves correspond to fits according to the phenomenological description of absorption saturation (see text for details).

**Figure 5 nanomaterials-13-01207-f005:**
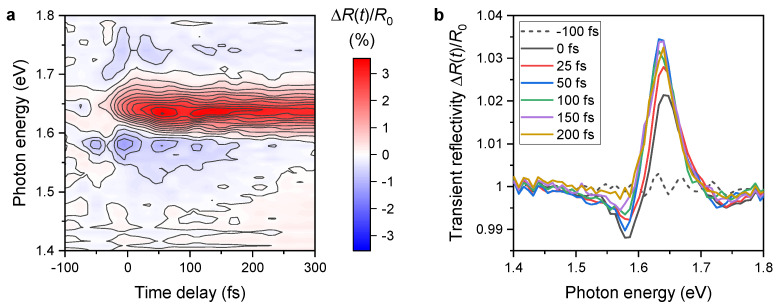
(**a**) Ultrafast dynamics of transient reflectivity of WSe${}_{2}$ monolayer measured with linearly polarized pump and probe pulses with crossed polarizations. (**b**) Transient reflectivity spectra in different time delays between the pump and probe pulses.

## Data Availability

The data that support the plots in this study are available from the corresponding author upon reasonable request.

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
