# Peer review of "Ultrafast Dynamics of Valley-Polarized Excitons in WSe_2_ Monolayer Studied by Few-Cycle Laser Pulses"

_nanomaterials, 2023, doi:10.3390/nano13071207_

Round 1
Reviewer 1 Report
In the work the authors present an investigation of spin-valley coupling in WS2 by means of ultrafast pump probe reflectivity. They use a few-fs laser source to investigate the early exciton dynamics in the 2D materials under circularly polarized pump-probe.
The paper is of interest for researchers working in the area of 2D materials, but several points must be addressed:
1. It is not clear what is the temporal resolution of the experimental setup. The title claims few-cycle pulses. Can the authors show the spectral phase interferometry measurements cited in line 49?
2. The authors must better explain the novelty with respect the existing current literature.
3. More details about the sample preparation must be added: What do they refer for gel film assisted mechanical exfoliation? Do they refer to the mechanical exfoliation of flakes or to the deterministic transfer of flake to flake? Can the author add relevant references? Also, what are the dimension of the investigated flakes with respect to the beam spot size?
4. More details about the setup must be added. The authors work at 75 MHz. How do the transient reflectance points are recorded? Do they detect the full spectra at one time or do they collect the single wavelength points? Do they use lock-in detection to measure the pump ON- pump OFF difference? Why do they use two choppers?
5. What is the probe wavelength of the transient dynamics shown in figure 3? Can the authors rescale the y-axis to better evidence the increase in the differential reflectance
6. The figure 3 and 4 show significative differences in the maxima of transient reflectance with respect the pump fluence while the authors say “The initial fast dynamics, which does not depend on pump fluence…” in line 127. Can the author better explain?
7. Can the author add a figure or an inset showing the two lifetimes in terms of the pump fluence? Is there a variation of the slower one with respect the pump fluence and how do they explain it?
8. Can the author give an estimate of the number of absorbed photons instead of just report the data in terms of pump fluence? The faster dynamics can be explained by the fact that a regime of more than one pump photon can lead to generation of more than one exciton. Are the measurements taken in regime of more or less than one absorbed photon?
Reviewer 2 Report
The paper by Koutensky et al. report on ultrafast dynamics of excitonic valley-polarization in WSe2 monolayer by using fs pump-probe reflection spectroscopy. The reflection signals exhibited two characteristic decays where the faster time constant of ~120 fs and the slower one of ~9 ps. The latter value decreased as the pump fluence increased.The similar results are obtained with the opposite polarization combinations, namely the σ- pump and σ+/σ- probe ones, confirming that the signal behaviors are not a possible influence of the experimental setup. These temporal behaviors are discussed in terms of the valley pseudo-spin formalism describing exitons' occupations in K+ and K- valleys prepared by the σ+ pump pulse, and the fast and slow decays are attributed to direct recombination of excitons and to valley depolarization induced by Coulomb exchange interactions, respectively. The experiments are well conducted and their interpretations are quite reasonable. Thus, I recommend it for publication in Nanomaterials.I only have a few minor comments that the authors would consider.
1) Provide schematic of band structure and optical selection rules for WSe2, for readers' convienience.
2) False color scale in Fig.2a should be reversed as in Fig.5a.
varies
3) Define f(q) in line 222.
Reviewer 3 Report
In this article the authors studied by reflection spectroscopy at room temperature in a monolayer sample of WSe2 onto SiO2/Si substrate and covered by a h-BN flake. The authors focused their study in the temporal evolution of the transient reflectivity spectrally integrated over the exciton resonance. The main finding is to obtain a fast decay time of (116 +/- 7 )fs they attribute to the initial relaxation dynamics. On the other hand they obtain a slower decay they atribute to the reduction of the exciton population.
I think their results could merit be published in Nanomaterials but the authors previously should enlarge and clarify the information about the sample fabrication and address a few other points:
1. Line 66 ".. are prepared by gel-film assisted mechanical exfoliation from" The authors should specify the gel-film they are using PDMS? Usually raman spectroscopy gives information about the quality of the exfoliated monolayer. The authors should clarify how they confirm that obtained monolayer flakes and information about the flake quality.
2. Line 68- "bulk crystalline silicon and 90 nm think layer of SiO2" The silicon crystalline wafer is intrinsic or doped? If doped what type of dopants?
3. Line 69. "multilayered hBN to protect them" Information about the hBN crystals they are using (quality, etc) and thickness of the multilayer flake of hBN are needed to be included.
4. Line 92 - As the author comment the substrate could have an important role in the measurements. So, why not using two layers of hBN to encapsulate the WSe2 monolayer? How the authors envisage can affect to use a graphite flake between the SiO2 and a back hBN to their results?
5. I wonder if the authors considered to use transmitivity by using a transparent substrate as PDMS or PMMA? with an encapsulated sample.
6. Line 141 - Low-temperature used to modify the exciton dynamics. The authors compare their time constant at RT with the coherence time obtained in Ref 19 at 10 K . The authors should comment why this comparison has sense and if both time scales are looking the same physics effect.
7. The combination of low-temperature and fully encapsulated monolayers of MoSe2 has revealed features that could not be observed at previous samples onto SiO2. See for example the work of Vaquero et al. Nanomaterials 2022, 12(3), 322; https://doi.org/10.3390/nano12030322. The authors should comment if a similar scenario it is expected for WSe2..
7. Photogating also could be playing a role in WSe2 monolayers as has been proved does in MoS2 monolayers (see for example D. Vaquero et al. Nanoscale 2021 DOI: 10.1039/d1nr03896). This effect could affect your results and in my opinion such a possiblity should be comment.
Reviewer 4 Report
In this work, the authors experimentally investigated the ultrafast dynamics of valley-polarized excitons in monolayer WSe2 using transient reflection spectroscopy with few-cycle laser pulses. They find that the anisotropic valley population of excitons decays on two different timescales at room temperature, and the mechanism is proposed. Generaly, the paper reports an interesting and innovative work. I recommend that the manuscript can be accepted after a major revision addressed the following comments.
1. As an experimental study, the conventional characterizations of the WSe2 are essential, i.e., SEM, Raman, AFM, PL.
2. In Line 128, “The initial fast dynamics, which does not depend on pump fluence, is probably related to the time in which the excitons are formed……”. The authors should give more discussions.
3. The valley-polarization of excitons in monolayer TMDs are significantly suppressed by the temperature. So, how is the anisotropic valley population of excitons under a lower temperature?
4. In Fig. 3, it is better to add the fitting curves.
5. The connection between the exciton occupation and pseudo-spin vector should give more attentions.
6. In line 214, it might be better to defining qT as the momentum of exciton in Brillouin zone rather than the momentum induced by thermal effect.
7. Should the “exciton mass” be corrected as “exciton effective mass”? If so, what is the specific value?
Round 2
Reviewer 1 Report
In their revised version the authors have addressed all the raised points. The paper can be accepted in present form
Author Response
The Referee wrote: "In their revised version the authors have addressed all the raised points. The paper can be accepted in present form."
We thank the Referee for this conclusion.
Reviewer 3 Report
The authors has addressed correctly many of my previous questions.
I keep my concerns about about the information included in the paper describing the sample fabrication.
I think the authors could improve their answers to my previous questions 1, 3 and 4 and include these data in this paper.
The aim of those questions was to help to enlarge the scope of the paper, with basic information to help others to reproduce their results like the type of gel-film used in the exfoliation, or the h-BN thikness of the cap layer. I do not pretend they fabricate new samples or make new experiments but they could discuss what they expect if they isolate better their sample from enviroment and substrate.
I strongly recommend the authors consider again to do so.
Please consider this comments as an optional minor revision previously to be published in nanomaterials.
Author Response
The Referee wrote: "
The authors has addressed correctly many of my previous questions.
I keep my concerns about about the information included in the paper describing the sample fabrication.
I think the authors could improve their answers to my previous questions 1, 3 and 4 and include these data in this paper.
The aim of those questions was to help to enlarge the scope of the paper, with basic information to help others to reproduce their results like the type of gel-film used in the exfoliation, or the h-BN thikness of the cap layer. I do not pretend they fabricate new samples or make new experiments but they could discuss what they expect if they isolate better their sample from enviroment and substrate.
I strongly recommend the authors consider again to do so.
Please consider these comments as an optional minor revision previously to be published in nanomaterials. "
We took into account the recommendations of Referee 3, revised the description of the sample preparation in the manuscript, and modified the corresponding part of the manuscript. The aforementioned changes are presented in the red-colored text in the revised version of the manuscript.
"
Round 3
Reviewer 3 Report
I thank the authors reconsider my suggestions. I regret they don't include the type of gel-film or the hBN thickness for their samples.